# Etiology of Pneumoparotid: A Systematic Review

**DOI:** 10.3390/jcm12010144

**Published:** 2022-12-24

**Authors:** Kazuya Yoshida

**Affiliations:** Department of Oral and Maxillofacial Surgery, National Hospital Organization, Kyoto Medical Center, 1-1 Mukaihata-cho, Fukakusa, Fushimi-ku, Kyoto 612-8555, Japan; omdystonia@gmail.com; Tel.: +81-75-641-9161

**Keywords:** pneumoparotid, systematic review, pneumoparotitis, parotid gland, etiology, intraoral pressure, Stensen’s duct

## Abstract

Pneumoparotid describes retrogradely insufflated air within the Stensen’s duct and/or parotid gland. It is a rare condition with variable causative factors. This study aimed to elucidate the clinical characteristics of pneumoparotid. Reports in all languages were evaluated following the Preferred Reporting Items for Systematic Reviews and Meta-Analyses statement 2020. A literature search was conducted using electronic medical databases (PubMed, Scopus, Web of Science, EBSCO, Ovid, Google Scholar, SciElo, LILIACS, and others) from 1890 to 30 June 2022. One hundred and seventy patients (mean age; 28.4 years) from 126 studies were reviewed. Common symptoms included swelling (84.7%) and pain (35.9%). Characteristic findings were crepitus in the parotid region (40%) and frothy saliva from the orifice (39.4%). The common etiologies included abnormal habits such as blowing out the cheeks (23.5%), idiopathic (20%), self-induced (15.9%), playing wind instruments such as trumpets or flutes (8.8%), and diseases inducing coughing or sneezing (8.2%). The treatments included antibiotic therapy (30%), behavioral therapy to avoid continuing causative habits (25.9%), psychiatric therapy (8.2%), and surgical procedures (8.2%). Treatment should be individualized and etiology-based. However, the etiology was not identified in 20% of patients. Further detailed data from larger samples are required to clarify and improve the recognition of this entity.

## 1. Introduction

Parotid region swelling is a relatively common complaint in otolaryngology, internal medicine, oral and maxillofacial surgery, and dentistry. Pneumoparotid is a rare cause of parotid swelling, which refers to the reflux of air into the parotid gland or Stensen’s ducts [1,2,3,4,5,6,7,8,9,10,11,12,13,14,15,16,17,18,19,20,21,22,23,24,25,26,27,28,29,30,31,32,33,34,35,36,37,38,39,40,41,42,43,44,45,46,47,48,49,50,51,52,53,54,55,56,57,58,59,60,61,62,63,64,65,66,67,68,69,70,71,72,73,74,75,76,77,78,79,80,81,82,83,84,85,86,87,88,89,90,91,92,93,94,95,96,97,98,99,100,101,102,103,104,105,106,107,108,109,110,111,112,113,114,115,116,117,118,119,120,121,122,123,124,125,126] (Appendix A). It is termed “pneumoparotitis” when it coexists with inflammation or infection. Hyrtl first described pneumoparotid in 1865 in wind instrument players [127]. Various English-language terms have been used for this condition, such as pneumoparotitis [15,17,21,25,26,27,28,29,31,32,33,34,36,37,39,45,48,49,53,54,57,59,61,63,64,66,67,68,72,78,82,83,84,86,88,90,95,100,103,104,106,109], pneumoparotiditis [12], pneumoparotis [29,64,101], pneumosialadentitis [24], pneumatocele glandulae parotitis [9], wind parotitis [13].

Pneumoparotid is an occupational hazard for glass blowers [1,2,3], wind instrument players [1,8,15,54], and divers or watchkeepers in a high barometric pressure environment [15]. Until the first half of the 20th century, pneumoparotid was recognized as a typical occupational disease in glass blowers [1,2,3,128,129,130]. In 1918, Trémollieres and Caussade [131] first reported self-induced pneumoparotid disease among the soldiers of a Moroccan tirailleurs regimen who exhibited parotid swelling on blowing into a small bottle, simulating mumps to avoid duty. The soldiers inflated their cheeks forcibly by pinching their nares and placing their hands over their mouths [131]. The episode was introduced as “factitious mumps” [132] or “simulation of mumps” [133,134]. Other etiologies of pneumoparotid vary widely. They include self-induction for various secondary gains, unconscious habits, diseases inducing coughing, sneezing, or vomiting, iatrogenic causes such as dental treatment, continuous positive pressure, spirometry, and unknown etiologies. Acute postoperative sialadenitis, termed “anesthesia mumps”, has been associated with various surgical procedures [135]. However, the majority of reports did not specify whether the cause of swelling was salivary stagnation or reflux of air [136]. Pneumoparotid is usually benign, and the swelling resolves spontaneously without any treatment.

A recent review on pneumoparotid and pneumoparotitis evaluated 54 patients from 49 English reports from 1987 to 2019 [120]. Due to this condition’s rarity, many cases might not have been correctly diagnosed and adequately treated. Therefore, identifying this entity is necessary through a thorough understanding of previously reported cases. On rare conditions such as pneumoparotid, data should be collected from as many cases as possible. Therefore, this study assessed and reviewed all reports in all languages on pneumoparotid, including clinical presentations and treatments.

## 2. Materials and Methods

### 2.1. Review of Literature

A systematic review was conducted according to the 2020 Preferred Reporting Items for Systematic Reviews and Meta-Analyses (PRISMA) statement criteria [137]. The literature search strategy was based on comprehensive electronic medical literature databases (PubMed, Scopus, Web of Science, EBSCO, Ovid, Google Scholar, SciELO, LILIACS, Japan Medical Abstracts Society, CiNii, and J-Stage) for the keywords “pneumoparotid,” “pneumoparotitis”, “pneumoparotitis”, and “pneumoparotide”. Furthermore, a manual search was conducted for articles on pneumoparotid or pneumoparotitis. Reports from 1890 until 30 June 2022, identified in these databases or via a manual search were screened by the author, with no language restrictions. Duplicate reports and double publications from the same cases were excluded. Exclusion criteria were records with missing fundamental information, such as sex, age, etiology, and clinical presentation, and those irrelevant to the purpose of this study., Further, all reports were assessed for eligibility and reviewed by the author. Reports on the “puffed-cheek” maneuver during computed tomography (CT) were excluded as they were irrelevant to the study objective. Reports of anesthesia mumps were included only when air was confirmed in the swelling. No restrictions were imposed with respect to the original text type.

### 2.2. Analysis

Fundamental clinical data were evaluated, including sex, age, etiology, clinical presentation, treatment, resolution, relapse, and follow-up. The disease duration was defined as the time from the onset of pneumoparotid to the first visit reported in each study. After a careful review of the reports, patients with pneumoparotid or pneumoparotitis were classified based on their etiologies, which were categorized into nine categories (abnormal habits, idiopathic, self-induced, diseases, wind instruments, glass blowing, iatrogenic, balloon, and others). Each etiology’s prevalence, mean age, and change over time were analyzed. If abnormal habits were considered intentional, they were classified as self-induced. Conditions that induce causative factors, such as persistent coughing, vomiting, blowing, or involuntary movements, were categorized as diseases. Cases with unidentified causative factors were classified as idiopathic. Differences in clinical data among the groups were statistically compared using a one-way analysis of variance. The Bonferroni method was used to conduct a post hoc test when an analysis of variance revealed significant differences. All analyses were conducted using the statistical software package SPSS for Windows (version 24.0; SPSS Japan Inc. Tokyo, Japan). The null hypothesis would be rejected at a 5% significance level (*p* < 0.05). Even if a study’s authors considered the cause, the case was classified as idiopathic if the cause was not direct. For example, chronic parotitis could be the inducing or predisposing factor of pneumoparotid disease; it could not be a direct cause unless other coexistent causes associated with an increase in intraoral pressure, were coexistent.

## 3. Results

The number of reports screened, assessed for eligibility, and included in the systematic review is presented in the flow diagram (Figure 1). The total number of records from the databases was 832 (PubMed; 98, Scopus; 83, Web of Science; 63, EBSCO; 215, Ovid; 34, Google Scholar; 258, SciELO; 4, LILIACS; 4, Japan Medical Abstracts Society; 30, CiNii; 36, and J-Stage; 7). Fifty-six records were obtained by manual searches of relevant papers or books. The search yielded 126 articles. Appendix A summarizes the results of the 126 reports. All studies were case reports or series. The languages and number of the evaluated literature were as follows: English: 80, Japanese; 13: German; 10: Korean; 8: Spanish; 6: French; 5: Italian; 1: Hebrew; 1: Chinese; 1: Turkish; 1. The number of cases was as follows: 112, two cases; 6, three cases; 2, four cases; 3, six cases; one, eight cases; and one, 14 cases.

### 3.1. Symptoms and Diagnoses

The 126 articles contained 170 patients (mean age; 28.4 ± 18.1 [standard deviation] years, range; 3–67 years). Forty-one were women (24.1%), and 124 were men (72.9%) (Appendix A). The disease duration was 24.8 ± 47.6 months (range 0–25 years) in 128 (74.9%) cases. The results of the symptoms and diagnostic imaging are summarized in Table 1.

Fifty-nine (34.7%) cases were bilaterally affected, 56 (32.9%) were left-sided, and 47 (27.6%) were right-sided. The most common symptom was swelling, observed in 144 (84.7%) cases, followed by pain in 61 (35.9%). Crepitus or classical cracking sensation in the parotid region (Figure 2A) was observed in 68 (40.0%) patients but not in 18 (10.6%) patients. Bubbles or frothy saliva from the orifice of Stensen’s duct were detected in 67 (39.4%) cases (Figure 2B) and were absent in 33 (19.4%) cases. The most severe inflammatory complication was emphysema. Among 30 (17.6%) cases, 23 (13.5%) were in the neck, and 6 (3.5%) were in the mediastinum.

CT was the most frequently used diagnostic imaging modality in 95 (55.9%) cases (Figure 3), followed by ultrasound in 47 (27.6%), sialography in 35 (20.6%), radiography in 29 (17.1%), sialendoscopy in 13 (7.6%), magnetic resonance imaging in 6 (3.5%), and fluoroscopy in 1 (0.6%). The typical imaging findings included air in the gland (100 [58.8%]) or duct (55 [32.4%]) and enlargement of the duct (27 [15.9%]) or gland (11 [6.5%]) (Table 2).

### 3.2. Etiology

Table 2 shows the details of each etiology. Most patients with abnormal habits were used to blowing out of the cheeks (Appendix A). While other common practices were puffing out of the cheeks to prevent irritation by orthodontic braces [37,72], biting the lower lip and whistling with a high-frequency sound [45], blowing out the cheek to stop aphthous ulcer-related pain while eating [19], and puffing the cheek during unbearable itching [49]. Thirty-four idiopathic patients did not play wind instruments, had no unconscious abnormal habits, and no other causative factors were detected. For various secondary gains, patients who were self-induced blew out the cheeks with a closed mouth, similar to the Valsalva maneuver (Appendix A). One patient reportedly self-injured the duct with pins [16]. The reasons for patients’ self-induced pneumoparotid were conflicts with parents [7,96], as an excuse for not going to school [11,40,48,96], an adjustment reaction to adolescence [12,17,21], to gain attention [69], mental disorders [16,39,56,96], to escape duty [67,131], and to leave prison [111]. Causative wind instruments, including toy instruments, were trumpets [5,13,54], flutes [66,76], a clarinet [13], a horn [15], a tuba [74], a recorder [77], fanfare [64], and a paper trumpet [104]. Diseases inducing pneumoparotid included coughing attacks [18,38,51,96], nervous tic [6,32,55], obstructive sleep apnea syndrome [42,125], sneezing crisis [10], mental disorder [17], clearing nares during hay fever attack [20], head and maxillofacial trauma [46], and vomiting [100]. Iatrogenic causes were dental air syringe [28,29,92,117], continuous positive airway pressure [108], air powder prophylaxis unit [33], spirometry [50], general anesthesia [85], upper endoscopy [91], and non-invasive positive pressure ventilation [102]. Most balloon blowers were children [13,42,76,94,98,122], whereas two were adults who blew for their children [79,97]. Other etiologies included decompression after diving [15], watchkeeping in a compartment [15], massage in the periauricular region [26], lifting heavy luggage [71], diving while retaining air in the oral cavity [76], facial trauma [87], diving while holding air in the oral cavity [93], and radiation therapy 126]. All glass blowers and balloon blowers were men. 

The percentage of each etiology in all patients is shown in Figure 4. The etiologies included abnormal habits (*n* = 39), idiopathic (*n* = 34), self-induced (*n* = 28), wind instruments (*n* = 15), diseases (*n* = 14), glass blowing (*n* = 12), iatrogenic (*n* = 11), balloon (*n* = 9), and others (*n* = 8). 

Figure 5 shows a comparison of the mean age between etiologies. The mean ages were as follows, abnormal habits; 28.4 ± 18.5 years, idiopathic; 37.4 ± 19.8, self-induced; 15.7 ± 9.3, diseases; 29 ± 17.7, wind instruments; 21.8 ± 16.5, glass blowing; 35.4 ± 10.1, iatrogenic; 43.5 ± 12.3, balloon;13.6 ± 10.8, and others; 23.2 ± 17.3. The mean age of each etiology differed significantly (*p* < 0.001, one-way analysis of variance). Self-induced patients (15.7 ± 9.3 years) were significantly younger than iatrogenic patients (43.5 ± 12.3 years, *p* < 0.001), idiopathic patients (32.4 ± 19.3 years, *p* < 0.01), and glass blowers (35.4 ± 10.1 years, *p* < 0.05) (Figure 5). Balloon blowers (13.6 ± 10.8 years) were significantly (*p* < 0.005) younger than iatrogenic patients (43.5 ± 12.3 years) and idiopathic patients (37.4 ± 19.8 years, *p* < 0.005).

Figure 6 shows the change in number of cases for each etiology since 1890. The number of reported cases has increased. The causes of the disease have drastically changed over time. Glass-blowing was the predominant cause in the early days; however, it completely disappeared after the latter half of the 20th century. Conversely, iatrogenic cases have increased since the 1990s. In addition, cases of abnormal habits and idiopathic causes are growing rapidly. Cases related to self-induction, diseases, wind instruments, and balloons are found regardless of time.

### 3.3. Treatments and Sequelae

The treatment outcomes and sequelae are shown in Table 3. Antibiotics were prescribed to 51 (30%) patients for anti-inflammatory treatment, often with steroidal anti-inflammatory drugs (Appendix A). Behavioral therapy, including explanation and instruction to refrain from abnormal habits, was provided to 44 (25.9%) patients (Appendix A). Psychiatric treatment was required in 14 patients [11,12,15,17,39,40,48,56,57,67,68,73,96,120]. The surgical procedures included aspiration [1,26,39,46,66,104,110,121], parotidectomy [14,15,40,90], rerouting [4,24,25,61], incision [8,39], ductal ligation [58,71], cystectomy [59,112], and irrigation [95,115]. In addition, continuous positive airway pressure (CPAP) [102,108] and oral appliances [125] were applied to patients with pneumoparotid related to sleep apnea syndrome. No information concerning treatment was found in 45 (27.1 %) patients.

Eighty-seven patients (51.2 %) were successfully treated (Appendix A) during an average of 9.9 ± 26.5 days. Ten (5.9%) patients’ conditions remained unresolved, and no data were described in 73 (42.9%). Relapse was reported in 29 (17.1%) patients, whereas 47 (27.6%) did not relapse, and no description of relapse was provided for 94 (55.3%). Follow-up was reported in 42 (24.7%) cases (20 ± 27.4 months), lost to follow-up in 7 (4.1%) cases, and not reported in 121 (71.2%) cases.

## 4. Discussion

To our knowledge, this is the first systematic review of all previously published reports in all languages, which were accessible through a database and manual search for the term pneumoparotid. Thus, this review was able to analyze more than three times as many patients as a recent review [120]. As a result, it became possible to classify the etiology of pneumoparotid in more detail due to the large number of cases, and new findings on this condition, such as the change in the number of patients with each etiology to date and the significant difference in mean age, became clear.

### 4.1. Protective Mechanism of the Orifice of the Stensen’s Duct

Several causes are associated with parotid gland swelling, such as acute infection (suppurative parotitis), chronic-specific infection (tuberculosis, sarcoidosis), viral infection (epidemic parotitis [mumps], Coxsackie A [human immunodeficiency virus]), sialoliths, sialadenosis, benign or malignant tumors (Warthin tumor, mucosa-associated lymphoid tissue), autoimmune diseases (Sjögren’s syndrome, Mikulicz’s disease), and endocrine disorders [138,139]. Pneumoparotid is a rare cause of parotid swelling. 

The normal anatomy of the Stensen’s duct protects air and saliva from the duct into the parotid glands in three stages: a much smaller diameter of the orifice than that of the duct, slit-shaped redundant mucous membrane folds, and increased angulation of the duct by distension of the cheek [12]. First, the diameter of the duct ostium is narrower than that of the duct. The average duct dimensions are 5 cm long and 3 mm wide [140]. The mean diameter of the Stensen’s duct at four different points along its length ranges between 0.5 and 1.4 mm [141]. The narrowing at the middle of the duct is striking, and the minimum width of the duct is located at the orifice [141]. Second, the duct orifice is slit-shaped with a redundant mucous membrane. It covers the orifice during increased intraoral pressure. The submucosal passage of the duct serves as a valvular mechanism, preventing inflation of the gland with increased intraoral pressure [140]. Third, the duct is compressed laterally along the masseter muscle and penetrates the buccinator muscle. Distention of the cheek increases the angulation of the duct, where it turns medially at the masseter muscle to pierce the buccal fat pad and is compressed by the buccinator muscles [12]. Consequently, beginner wind instrument players who blow with “full cheeks” are more likely to have pneumoparotid than those who blow using an adequate embouchure technique with “contracted cheeks” [127]. 

Amano et al. [142] studied the relationship between the parotid duct and buccal muscle and investigated the structures around the orifice via scanning electron microscopy. The duct showed circular mucoepithelial rugae on the inner luminal surface before entering the buccal muscle. After entering the muscle, longitudinal rugae were observed on the inner luminal surface, and after entering the buccal submucosal tissue, a flat, torus-like morphology was observed. In some cases, the orifice of the parotid duct showed a drawstring purse-shaped morphology owing to the longitudinal torus. The morphological features of the parotid duct adjust the salivary flow, preventing the countercurrent of the liquid [142]. Furthermore, Amano et al. [143] suggested that the area of the duct penetrating the buccinator muscle plays a role in regulating salivary passage through the contraction of the surrounding buccinator muscle fibers.

### 4.2. Pathology of Pneumoparotid

If intraoral pressure exceeds the protective mechanism, air can reflux through the duct into the parotid gland, leading to pneumoparotid. Donders [144] reported that the normal tracheal pressure is at most 2–3 mmHg during expiration. Scheier [3] measured 150 mmHg of increased intraoral pressure during glass blowing. Intraoral pressures were recorded using various wind instruments [145,146,147]; in most wind instruments, it increases with pitch and loudness and varies from 2.5 to 158 mmHg [145]. The average values of the highest maximum pressures are 134 mmHg on trumpet and 170 mmHg on a piccolo trumpet [146]. A high-speed dental air turbine can produce an intraoral pressure of 2.2 kPa (16.5 mmHg) [29]. Air-powder prophylactic cleaning units create an intraoral pressure of 55–60 psi (60–65 mmHg) [33]. The Valsalva maneuver involves expiratory effort against a closed mouth and/or glottis in the sitting or supine position with increased intraoral and intrathoracic pressure raised to 40 mmHg for 15–20 s [148]. The Valsalva maneuver is used intraoperatively for diagnostic and therapeutic purposes during specific surgical procedures [148]. However, if patients with self-induced pneumoparotid perform this maneuver by forcefully attempting exhalation with a closed mouth and while pinching the nose shut, the intraoral pressure may increase drastically. If high intraductal pressure continues or the injuries are repeated, the parotid acini may rupture. Air may enter through the ruptured acini or ducts, entering the parapharyngeal space and causing cervical subcutaneous emphysema. Crepitation is the principal sign of emphysema, indicating rupture of the gland capsule. The retropharyngeal space may be involved in further progression, resulting in pneumomediastinum or pneumothorax. Emphysema may be observed in the face (17.6%), neck (13.5%), and mediastinum (3.5%) (Table 2).

Acute postoperative parotid gland swelling can be observed in association with general anesthesia, called “anesthesia mumps” [135], and has also been reported in different surgical procedures, “surgical mumps” [149]. Mundé first observed it in 1878 following ovariotomy [150]. Although the precise mechanism of the disease remains unclear, some predisposing factors have been postulated. For example, straining, coughing, and sneezing during general anesthesia or post-anesthesia can increase intraoral pressure. In addition, agents such as succinylcholine, as muscle relaxants, cause the loss of buccinator muscle tone around the orifice of Stensen’s duct [135], which facilitates the forced reflux of air into the duct. Furthermore, the activation of the pharyngeal reflex causes parasympathetic nerve stimulation, which results in vasodilation and hyperemia of the parotid gland [151,152]. Along with coughing or straining the endotracheal tube, the violence of endotracheal intubation serves as a stimulus for the pharyngeal reflex [151,153,154]. The presence of air within the swelling can be confirmed by identifying crepitation and via diagnostic imaging. However, whether the cause of swelling is saliva stagnation or air reflux typically remains undiagnosed [136]. This report included only one case that Tekelioglu et al. [85] confirmed as pneumoparotid.

In addition, parotid swelling has been reported after endoscopy, bronchoscopy, and rigid esophagoscopy [152,155]. Particularly salivary gland swellings during peroral endoscopy were named “Compton’s pouches” [156,157]. The pouches disappear spontaneously in half an hour or up to several hours and are associated with no subjective symptoms except a soft swelling in the parotid region [156,158,159,160]. Palmer and Boyce [156] considered these pouches to be blind remnants of brachial clefts—probably the fourth pair—into which air has been forced due to straining. However, it is clinically unlikely for the remaining pouches of the branchial clefts to open in the oral cavity. It seems more reasonable to consider that forced air insufflates retrogradely from the salivary gland orifice in association with a rapid increase in intraoral pressure related to gagging and retching during peroral endoscopy [158,159,160]. Kuriyama [161] reported the case of a 52-year-old woman with painless bilateral parotid swelling after forcefully blowing a clogged drain hose of a washing machine. The swelling disappeared spontaneously the next day. He recognized the case as a type of Compton’s pouch; however, the patient had a pneumoparotid similar to that after balloon blowing.

### 4.3. Predisposing Factors of Pneumoparotid

The underlying pathophysiology of pneumoparotid appears to begin with abnormal dilation of the orifice or duct, which makes air reflux possible. Several predisposing factors for pneumoparotid disease have been reported in the literature. These factors include a patulous duct, congenital abnormality, hypotonia of the buccinator muscle around the papilla, hypertrophy of the masseter muscle, self-injury to the Stensen’s duct, decreased production of saliva with increased mucous secretion, transient obstruction of Stensen’s duct by mucous plugs causing decreased salivary flow, diagnostic maneuvers such as sialography [21,23,39,48,90], blunt trauma to the cheek [24], pleomorphic adenoma [36], and treatment for duct stenosis with stent implantation or transoral duct surgery [162]. Sánchez et al. [16] reported the case of a 15-year-old girl who injured herself by self-instrumentation of the parotid gland orifice with hairpins or safety pins. Furthermore, decreased salivary secretion may also be a predisposing factor. Three patients [108,113,126] had Sjögren syndrome, and two had xerostomia [15,96] in this review. 

As mentioned above, the contraction of the buccinator muscle serves as a protective mechanism for the duct. Blowing with “full cheeks” weakens this mechanism. Excessive stretching of the cheeks can result in the attenuation of buccinator muscle fibers. Hyperactivity of the masseter muscle has been postulated as a predisposing factor. Goncalves et al. [126] reported four out of 14 patients with pneumoparotid had bruxism. They hypothesized that increased intraoral cheek pressure could help overcome the protective mechanism of Stensen’s duct papilla and thus facilitate the occurrence of pneumoparotid. Masseter hypertrophy can be associated with compression of the parotid gland duct system, which may interfere with the normal salivary flow, producing parotid swelling and pain [163]. Goncalves et al. [126] postulated that bruxism could irritate the buccal mucosa, and buccal musculature may cause sphincter insufficiency, resulting in pneumoparotid. Further studies are required to confirm this hypothesis. Pneumoparotid can occur after treatment of duct stenosis with stent implantation and/or transoral duct surgery in the distal duct system [126]. 

### 4.4. Diagnosis of Pneumoparotid

With prior knowledge of the pneumoparotid gland, reaching a diagnosis is not difficult. However, pneumoparotid is often misdiagnosed as parotitis. Therefore, a detailed and careful medical history and confirmation of characteristic clinical signs (crepitus on the parotid gland and frothy saliva from the orifice) are needed at the first step during the examination. Therefore, it is necessary to avoid numerous negative studies to exclude other causes of parotid swelling. 

Air in the duct and/or gland can be diagnosed using a CT scan, which is readily available in most clinics. Some researchers have stated that CT is the gold standard for diagnosing pneumoparotid [30,48,88,90,120], as it visualizes the anatomy and a great deal of information, such as small amounts of air and the extension of the pathologic process in a short time. However, CT scans have disadvantages, such as radiation exposure and cost. Aghahei Lasboo et al. [164] confirmed the diagnosis of pneumoparotid using a “puffed-cheek” technique. CT was performed after sialography. The second CT scan confirmed a reduction in the air after massage. The patient was requested to puff out the cheeks to increase intraoral pressure, and a third CT scan confirmed the presence of air in Stensen’s and intraductal ducts. Although this is a reliable method, it is not necessary for clinical use from the viewpoint of radiation exposure and cost. 

Ultrasound showed multiple hyperechoic areas corresponding to air within the ducts and parenchyma of the gland. Therefore, ultrasonography is useful for diagnosis and follow-up [84,109]. Moreover, it is non-invasive, cost-effective, and provides real-time dynamic imaging. Recently, Goncalves et al. [126] reported detailed ultrasound examination findings in 21 patients with pneumoparotid (seven had secondary pneumoparotid following duct stenosis), concluding that ultrasound helps characterize the pneumoparotid and serves as an imaging tool. An experienced specialist in salivary gland disease could skillfully examine and diagnose pneumoparotid using ultrasound. However, testing with ultrasound may rarely be repeatable, requires skill, and is difficult for non-professionals. 

Sialography remains the standard method for demonstrating sialectasia, radiolucent calculi, duct strictures, and inflammatory diseases of the salivary gland duct system, including duct dilation starting in the parotid duct and the secondary ducts [60]. Sialendoscopy is a routine diagnostic and minimally invasive therapeutic procedure that aims to evaluate and manage salivary ductal system disorders, including chronic inflammatory conditions [95]. The use of sialendoscopy in the diagnosis and screening of pneumoparotitis has recently increased [87,95,108,113,126]. High-resolution magnetic resonance imaging using a surface coil and magnetic resonance sialography can be a beneficial diagnostic tool, which is painless and avoids radiation exposure [49]. However, it is expensive and unsuitable for children because of its long inspection time. Chest radiography should be performed when pneumomediastinum or pneumothorax is suspected.

### 4.5. Etiology of Pneumoparotid

Pneumoparotid was observed in 6–10% of glass blowers until the first half of the 20th century [3]. More pressure is necessary to produce larger glassware than delicate work and is more likely to cause pneumoparotid [2,3]. Therefore, pneumoparotid is recognized as an occupational disease in glass workers. 

Playing wind instruments, such as the trumpet [5,8,13,54], flute [66,76], horn [15], tuba [74], clarinet [13], or recorder [77] were also considered occupational hazards. Hyrtl [127] stated that if intraoral pressure exceeds the protective mechanism of Stensen’s duct when playing a wind instrument, air can pull in retrogradely from the orifice of the duct. Moreover, he suggested that beginners are more susceptible to air entry when they blow with their cheeks full and less likely when they learn the appropriate embouchure technique. Most patients were beginners; however, some were professionals [8,15,54]. Therefore, patients must learn not to blow with full cheeks and master embouchure techniques.

Gazia et al. [120] reviewed 49 reports and analyzed a total of 54 patients, reporting that the most frequent etiology was self-induction by swelling of the cheeks (53.7%). Nevertheless, whether the abnormal habit was intentionally induced pneumoparotid for personal gain or unconscious habit has a significant difference in etiology. Therefore, in this study, only intentional abnormal habits for secondary gain were classified as self-induced, and other unconscious responses were classified as abnormal habits. Most patients with abnormal habits blew air out of their cheeks. The other patients had a habit of puffing out the cheeks to prevent irritation by orthodontic braces [37,72], biting the lower lip and whistling with a high-frequency sound [45], blowing out the cheek to stop aphthous ulcer pain while eating [19], and puffing the cheek during unbearable itching [49]. Without distinguishing between abnormal habits and self-induction, as Gazia et al. did, the proportion of patients with self-induction was 39.4% in this review.

Self-induced patients blew out the cheeks with a closed mouth, similar to the Valsalva maneuver, for various secondary gains. For example, a 15-year-old female with psychological problems developed pneumoparotitis due to self-instrumentation of the gland with hairpins and safety pins [16]. The reasons for patients with self-induced pneumoparotid included conflicts with parents [7,96], an excuse for not going to school [11,40,48,96], an adjustment reaction to adolescence [12,17,21], and to obtain attention [69]. Self-induced patients, for such reasons, were adolescents with psychological problems or mental disorders [16,39,56,96]. However, some adults self-induced this entity to escape their duties [67,131] or leave prison [111]. Brasseur et al. [56] reported a 31-year-old female who presented with automultilation and intentional production of physical symptoms and was diagnosed with Munchausen syndrome. Ino et al. [96] reported eight patients with pneumoparotid syndrome and suggested that some may have had Munchausen syndrome. Munchausen syndrome is characterized by individuals who intentionally and deliberately produce signs and symptoms of a disease and tend to seek medical or surgical care [165,166]. The lack of identification of this condition may lead to unnecessary laboratory tests and procedures that may prolong hospitalization and increase the costs to health systems [167]. To date, no effective treatments have been demonstrated through well-conducted studies, and no diagnostic criteria exist [167].

No causative factors were identified in a relatively large number of patients (20%) in this study. They had no abnormal habits, playing wind instruments, iatrogenic episodes, or disease-causing coughing or vomiting. The average age of these patients (37.4 years) was higher than that of the other patients. A recent case study reported a 57-year-old man with pneumoparotid related to sleep apnea syndrome, who was treated effectively with an oral appliance with an anterior opening to reduce intraoral pressure [125]. The patient did not snore but blew overnight. Obstructive sleep apnea is characterized by repetitive, complete, or partial closure of the upper airway during sleep [168,169]. Therefore, CPAP therapy is an effective treatment option. However, oral appliances are an essential treatment choice and may be the preferred initial treatment for mild-to-moderate obstructive sleep apnea syndrome or snoring [168,169]. When CPAP cannot be properly regulated, it can increase intraoral pressure and induce pneumoparotid [108]. Some idiopathic cases in this review might have had sleep apnea syndrome.

The causative wind instruments, including toy instruments, include trumpets [5,13,54], flutes [66,76], clarinets [13], horns [15], tuba [74], recorders [77], fanfares [64], and paper trumpets [104]. Diseases inducing pneumoparotid disease include coughing attacks [18,38,51,96], nervous tic [6,32,55], obstructive sleep apnea syndrome [42,125], sneezing crisis [10], mental disorder [17], clearing nares during hay fever attack [20], head and maxillofacial trauma [46], and vomiting [100]. All of these causes are associated with an increase in intraoral pressure. 

The iatrogenic causes were dental air syringes [28,29,92,117], continuous positive airway pressure [108], air powder prophylaxis units [33], spirometry [50], general anesthesia [85], upper endoscopy [91], and non-invasive positive pressure ventilation [102]. Since dental syringes generate high pressure, attention should be paid to the direction of their use when treating maxillary molars. Likewise, pressure adjustment and follow-up are important for CPAP and non-invasive positive pressure ventilation. 

Other etiologies include decompression after diving [15], watchkeeping in a compartment [15], massage in the periauricular region [26], lifting heavy luggage [71], diving with air in the oral cavity [76], facial trauma [87], diving while holding breath [93], and radiation therapy [126]. Incidental asymptomatic pneumoparotid occurred after the CT with the “puffed-cheek” maneuver in two studies; however, these were not included because they were unsuitable for this review, and the patients’ basic information was not reported [170,171]. Ahuja et al. [170] reported that five of 80 patients had this condition after CT scans; all remained clinically asymptomatic. Bhat et al. [171] reported pneumoparotid in 47 out of 300 patients after multidetector CT with “puffed-cheek”. A few patients experienced transient fullness immediately after the procedure; however, no patient had lasting or infective symptoms. 

The mean age of the patients with each etiology was significantly different (Figure 5). Patients with self-induced pneumoparotid were mainly children with a low mean age. However, there were other reasons for self-induction, including to escape duty [67,131] or to leave prison [111]. Likewise, most patients related to balloon blowing were children, and the mean age was low. However, two adults blew balloons on their children [79,97]. Glass blowers are forced to work long hours in harsh environments for many years [1,4,130]. Therefore, their disease duration was long, and their average age was also high.

The cause of the condition has changed drastically over time (Figure 3). In the early days, glass blowing was the predominant cause; however, after the latter half of the 20th century, it was reduced entirely owing to subsequent innovations and mechanization in the glass industry. However, iatrogenic cases have increased since the 1990s. Iatrogenic factors include dental procedures [28,29,33,92,117], CPAP [108], spirometry [50], general anesthesia [85], upper endoscopy [91], and non-invasive positive pressure ventilation [102]. Moreover, cases with abnormal and idiopathic causes are increasing rapidly. The reason for the increase in pneumoparotid numbers due to abnormal habits remains unknown. However, a gradually growing awareness of pneumoparotid associated with abnormal habits may result from correctly recognizing cases previously diagnosed as recurrent parotitis. Cases related to self-induced diseases, wind instruments, and balloons can be found, regardless of the period.

### 4.6. Treatment of Pneumoparotid

A definite diagnosis of the etiology in each patient with pneumoparotid is indispensable and a premise of treatment or the prevention of complications. Some authors suggest that pneumoparotid should be divided into isolated acute events and recurrent injuries [40,70]. This division is relevant to the treatment method and the expected prognosis. In cases of incidental pneumoparotid, such as balloon blowing or complications of dental procedures, antibiotic prophylaxis may be required if there is a possibility of infection. Acute episodes should be managed conservatively with parotid stimulation techniques such as sialagogues, warm compresses, and massage [40,68,88,90].

Pneumoparotid due to wind instruments is observed mainly in beginners [127]. Instead of blowing full cheeks, patients need to learn proper playing embouchure techniques. However, some patients were professional players [8,15,54]. Even a famous trumpet artist showed attenuation or deformation of the buccinator muscle fibers resulting from excessive stretching of the cheeks due to repeated and heavy play of the trumpet [172]. Therefore, they should be retrained using the embouchure technique to decrease the likelihood of air ingress or advised to blow with “contracted cheeks”.

Intentionally self-induced patients should be treated with psychiatric methods such as counseling, pharmacotherapy, and cognitive behavior therapy. Supportive psychotherapy is necessary in some cases. Some patients with abnormal habits are unaware of the roles of these habits in the causation of pneumoparotid disease. Therefore, explanation and guidance to discontinue their habits are necessary for patients with abnormal habits. If they are ineffective, further counseling or psychiatric treatment is necessary. Cases suspected of being affected by psychiatric diseases require psychiatric consultation. 

Antibiotics and steroidal anti-inflammatory drugs are the most commonly used treatments. Anti-inflammatory treatment for acute inflammation should be prioritized in patients with emphysema. Cases of more extensive inflammation accompanying emphysema require emergency treatment. In contrast, patients with chronic parotid gland and duct symptoms require otorhinolaryngological surgical treatment. 

Surgery may be considered a last resort in recurrent or chronic refractory pneumoparotitis. Konstantinidis et al. [95] suggested sialendoscopy and irrigation with steroids as another treatment modality in cases of recurrent pneumoparotitis, avoiding major surgery. Han and Isaacson [58] described parotid duct ligation as an appealing first in the surgical treatment of recurrent pneumoparotid. The surgical transposition of the duct orifice to the tonsillar fossa via a submucosal tunnel [173] is to create an elongated path deep in the buccal mucosa that would compensate for the increased intraoral pressure, thereby decreasing the risk of auto-insufflation [24]. Patients may require parotidectomy because of noncompliance, treatment failure, or chronic infection [90]. The endpoint of recurrent pneumoparotitis is chronic parotitis, and its standard treatment is parotidectomy [174]. However, it carries the risk of facial nerve injury. 

Some clinicians have reported patients with pneumoparotid accompanied by obstructive sleep apnea syndrome [99,102,108]. Long-term use of oronasal CPAP [102,108] or mandibular advancement devices [99] is a potential cause of pneumoparotid. Cabello et al. [99] reported a patient with pneumoparotid disease who blew severely overnight. Various oral appliances have been increasingly used to treat sleep apnea syndrome [175,176,177]. Guidelines on sleep apnea syndrome treatment have extended the indications of the oral appliance to moderate and severe sleep apnea patients when a patient refuses CPAP therapy [178]. Many randomized trials have confirmed the effectiveness of oral appliances [179,180,181,182,183,184]. Oral appliances are principally constructed by advancing the position of the mandible to enlarge the upper airway dimensions and possibly affect upper airway muscle tone [185]. Conversely, in a recent study [125], a patient with pneumoparotid-accompanied sleep apnea did not snore but blew overnight. Therefore, suppression of events that increase intraoral pressure in such patients is paramount [125]. In this review, causative factors were not identified in 20% of the patients. The average age of the idiopathic cases was 37.4 years. Adults over the age of 30 years are unlikely to have intentionally self-induced pneumoparotid or conceal abnormal habits. Twenty-four percent of men aged 30–60 years had sleep-disordered breathing with an apnea-hypopnea index ≥5/h [186], and the overall population prevalence ranged from 9% to 38% and was higher in men [187]. Therefore, some cases must have an unknown etiology whose pneumoparotid is related to intraoral pressure increase due to sleep apnea syndrome. The relationship between pneumoparotid and sleep apnea syndrome remains to be elucidated. Although such a setting is difficult because of the rarity of idiopathic pneumoparotid, further studies confirmed by polysomnography with video recording in larger cases may be necessary to clarify this hypothesis.

### 4.7. Limitations and Future Directions

Since previous reports concerning pneumoparotid lacked available data, the author could not perform common analyses, such as synthesis of results or meta-analyses. In this report, to clarify the differences between the two categories, isolated acute events and recurrent injuries, the results were divided into two categories and statistically analyzed; however, data were missing, preventing a reliable statistical outcome. Authors of case reports or case series should provide fundamental data that are amenable to evaluation in later systematic reviews or meta-analyses.

Unfortunately, most of the publications analyzed in this report were not well described. Many reports did not mention the fundamental findings, such as sex, age, etiology, clinical presentation, treatment, resolution, and relapse. Of the papers evaluated in this study, 42.9% did not describe the resolution, and 55.3% and 71.3% did not mention relapse or follow-up, respectively. The citation method in the literature for the papers evaluated in this report was not sufficient. Many studies must have referred to previous reports without checking the original literature and often cited the description of previous reports without citing the original literature. For example, Hemphill [188] described an episode of self-induced pneumoparotid reported by Trémollieres and Caussade in 1918 [131]. Hemphill read information from an old English textbook. However, he did not cite the textbook as he had forgotten the book’s name [188]. The book must be “Survey of Head Surgery” [134]. Nevertheless, many case reports have repeatedly cited this paper without checking the original literature. Authors of scientific reports must be responsible for the accuracy of references in their publications.

All papers evaluated in this study were case reports or case series. Most of these (88.9%) were single case reports. Therefore, comparative studies between the treatment methods are lacking in the literature. The rapid increase in reports of pneumoparotid (Figure 4) suggests an increasing awareness of pneumoparotid. Since this entity is very rare, multicenter trials are needed. The cause or etiology in 20% of patients remains unknown. Further detailed data and larger samples are necessary to improve recognition and understanding of this entity, clarify unknown etiology, and explore further treatment choices or prevent complications.

## 5. Conclusions

The etiology of pneumoparotid is variable and goes unidentified in 20% of patients. Further detailed data from larger samples are necessary to better understand and improve the identification of this entity.

## Figures and Tables

**Figure 1 jcm-12-00144-f001:**
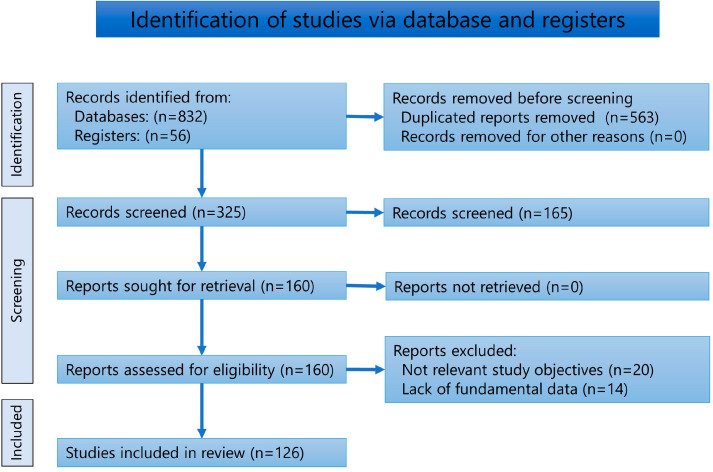
PRISMA diagram of the literature search and screening.

**Figure 2 jcm-12-00144-f002:**
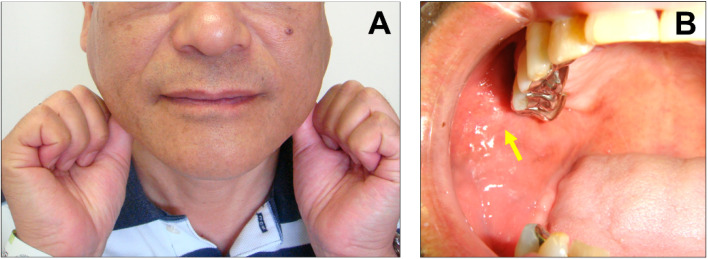
Typical symptoms of pneumoparotid. Crepitus on the parotid region (**A**) and frothy or bubbly saliva from the orifice (yellow arrow) (**B**).

**Figure 3 jcm-12-00144-f003:**
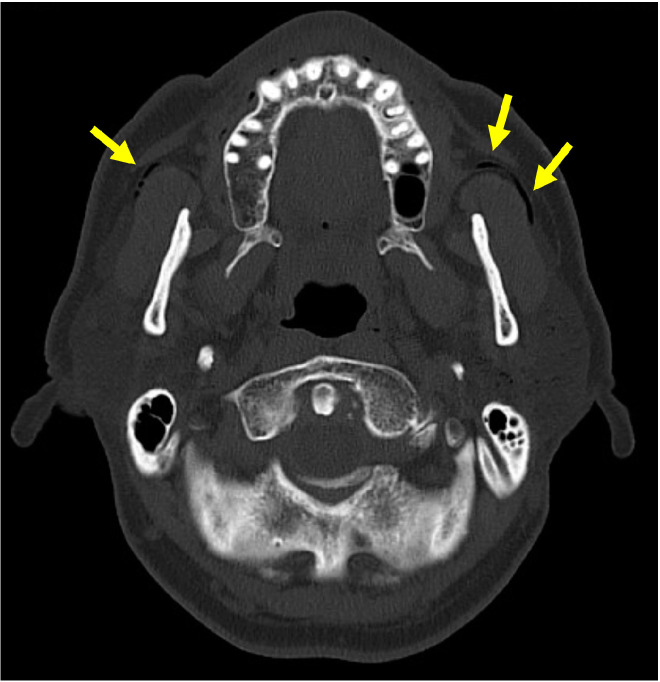
Air in the bilateral parotid glands and ducts (yellow arrows) in an axial-section computed tomography (CT) scan.

**Figure 4 jcm-12-00144-f004:**
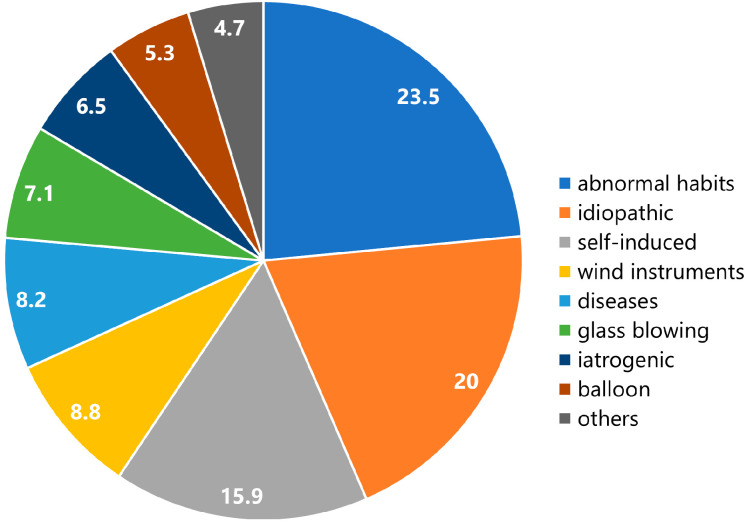
Etiologies of patients with pneumoparotid. The numbers in the pie chart sectors represent percentages.

**Figure 5 jcm-12-00144-f005:**
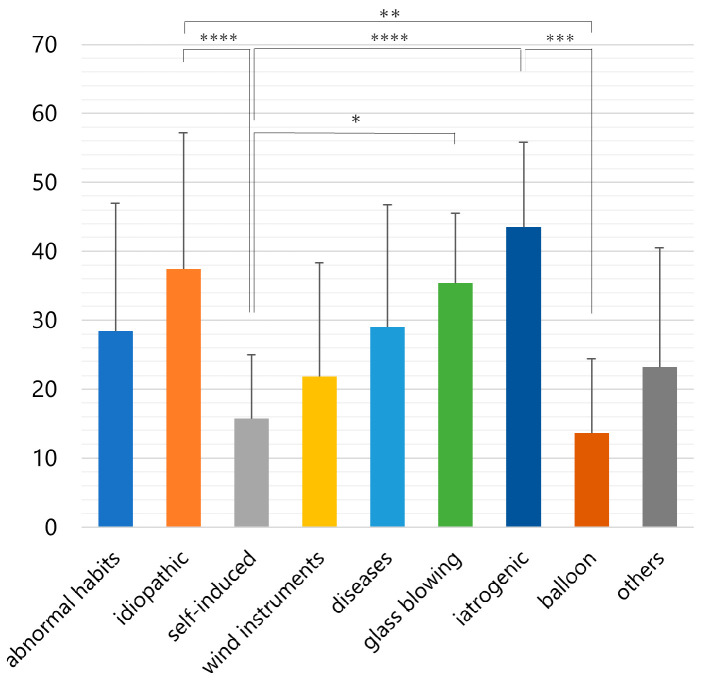
Mean age for each etiology. * *p* < 0.05, ** *p* < 0.01, **** p* < 0.005, ***** p* < 0.001.

**Figure 6 jcm-12-00144-f006:**
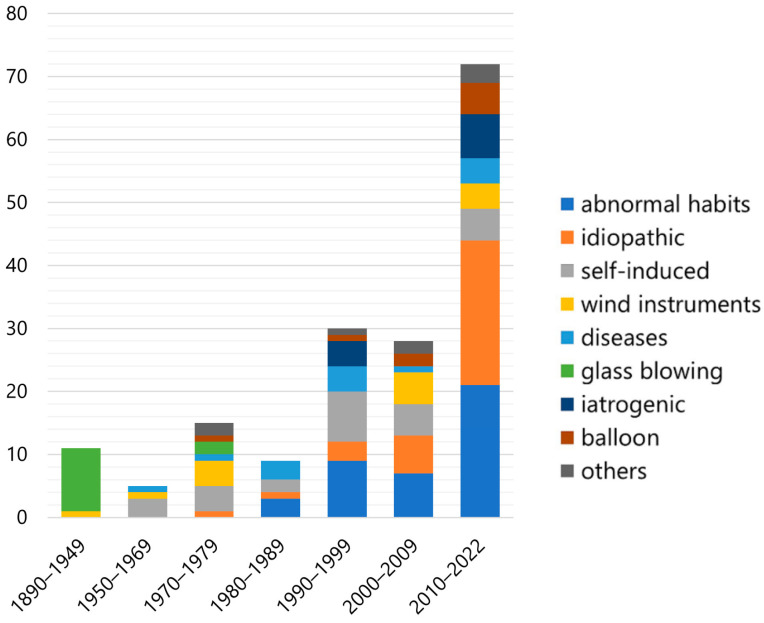
Changes in the number of patients for each etiology of pneumoparotid during the past 130 years.

**Table 1 jcm-12-00144-t001:** Symptoms and diagnostic images. NR; not reported, CT; computed tomography, US; ultrasound, MRI; magnetic resonance imaging.

Affected side, [*n* (%)]	Bilateral: 59 (34.7%), left: 56 (32.9%), right: 47 (27.6%), NR: 8 (4.7%)
Crepitus on the parotid region, [*n* (%)]	Yes: 68 (40%), No: 18 (10.6%), NR: 84 (49.4%)
Bubbles from the orifice of the duct, [*n* (%)]	Yes: 67 (39.4%), No: 33 (19.4%), NR: 70 (41.2%)
Symptoms, [*n* (%)]	Swelling: 144 (84.7%), pain: 61 (35.9%), discomfort: 8 (4.7%), noise 6 (3.5%)
Emphysema, [*n* (%)]	Face: 30 (17.6%), neck: 23 (13.5%), mediastinum: 6 (3.5%)
Diagnostic images, [*n* (%)]	CT: 95 (55.9%), US: 47 (27.6%), sialography: 35 (20.6%), radiography: 29 (17.1%), sialendoscopy: 13 (7.6%), MRI: 6 (3.5%), fluoroscopy: 1 (0.6%), NR: 26 (15.3%)
Imaging findings, [*n* (%)]	Air in the gland: 100: (58.8%), air in the duct: 55 (32.4%), enlarged duct: 27 (15.9%), emphysema: 19 (11.2%), mass: 14 (8.2%), enlarged gland: 11 (6.5%)

**Table 2 jcm-12-00144-t002:** Causes for each etiology. NR; not reported.

Abnormal habits, *n* = 40 women; 12, men; 28	Blowing out the cheeks; *n* = 35, puffing out of the cheeks to prevent irritation by orthodontic braces; *n* = 2, biting the lower lip and whistling with a high-frequency sound; *n* = 1, blowing out the cheek to stop aphthous ulcer pain while eating; *n* = 1, puffing the cheek during unbearable itching; *n* = 1
Idiopathic, *n* = 34 women; 8, men; 26	No causative factors were identified; *n* = 34
Self-induced, *n* = 27 women; 7, men; 20	Blowing out the cheeks with closed mouth; *n* = 26, self-injury into the Stensen’s duct with pins; *n* = 1
Wind instruments, *n* = 15 women; 4, men; 7, NR; 4	Trumpet; *n* = 4, flute; *n* = 2, clarinet; *n* = 1, horn; *n* = 1, tuba; *n* = 1, fanfare; *n* = 1, paper trumpet; *n* = 1, recorder; *n* = 1, NR; *n* = 3
Diseases, *n* = 14 women; 4, men; 9	Coughing attack; *n* = 4, nervous tic; *n* = 3; obstructive sleep apnea syndrome; *n* = 2, sneezing crisis; *n* = 1, mental disorder; *n* = 1, clearing nares during hay fever attack; n = 1, head and maxillofacial trauma; *n* = 1, vomiting; *n* = 1
Glass blowing, *n* = 12 women; 0, men; 12	Professional glass blower; *n* = 12
Iatrogenic, *n* = 11 women; 4, men; 6, NR; 1	Dental air syringe; *n* = 4, continuous positive airway pressure; *n* = 2, air powder prophylaxis unit; *n* = 1, spirometry; *n* = 1, general anesthesia; *n* = 1, upper endoscopy; *n* = 1, noninvasive positive pressure ventilation; *n* = 1
Balloon, *n* = 9 women; 0, men; 8, NR; 1	Balloon blowing; *n* = 7, balloon blowing for their children; *n* = 2
Others, *n* = 8 women; 2, men; 6	Decompression after diving; *n* = 1, watchkeeping in a compartment; *n* = 1, massage in the periauricular region; *n* = 1, lifting heavy luggage; *n* = 1, diving with air in the oral cavity; *n* = 1, facial trauma; *n* = 1, diving while holding breath; *n* = 1, radiation therapy; *n* = 1

**Table 3 jcm-12-00144-t003:** Treatment outcomes and sequelae. NR; not reported, SD; standard deviation.

Treatment, [*n* (%)]	Antibiotics; *n* = 51 (30%)
	Behavioral therapy; *n* = 44 (25.9%)
	Psychiatric therapy; *n* = 14 (8.2%)
	Analgesics; *n* = 10 (5.9%)
	Aspiration; *n* = 8 (4.7%)
	Massage; *n* = 4 (2.4%)
	Parotidecomy; *n* = 4 (2.4%)
	Rerouting; *n* = 4 (2.4%)
	Continuous positive airway pressure; *n* = 3 (1.8%)
	Incision; *n* = 3 (1.8%)
	Ductal ligation; *n* = 2 (1.2%)
	Irrigation; *n* = 2 (1.2%)
	Oral appliance; *n* = 1 (0.6%)
	NR; *n* = 45 (27.1%)
Resolution, [*n* (%), mean ± SD]	Yes; *n* = 87 (51.2%), 9.9 ± 26.5 (days), range (0–180),
	No; *n* = 10 (5.9%)
	NR; *n* = 73 (42.9%)
Relapse, [*n* (%)]	Yes, *n* = 29 (17.1%)
	No; *n* = 47 (27.6%)
	NR; *n* = 94 (55.3%)
Follow-up, (months) [*n* (%), mean ± SD]	Yes; n = 42 (24.7%), 20 ± 27.4, range (1 week–10 years),
	No; *n* = 7 (4.1%)
	NR; *n* = 121 (71.2%)

## Data Availability

Not applicable.

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
