# Peer review of "Etiology of Pneumoparotid: A Systematic Review"

_jcm, 2022, doi:10.3390/jcm12010144_

Round 1
Reviewer 1 Report
Introduction
- Too many references (line 33-34). I would eliminate all this “…such as pneumoparotitis [15, 17, 21, 25-29, 31-34, 36, 37, 39, 45, 48, 49, 33 53, 54, 57, 59, 61, 63, 64, 66-68, 72, 78, 82–84, 86, 88, 90, 95, 100, 103, 104, 106, 109], pneu- 34 moparotiditis [12], pneumoparotitis [29, 64, 101], pneumatozele [1, 4, 128], pneumosiala- 35 dentitis [24], pneumatocele glandulae parotitis [9], wind parotitis [13], pneumoparotidite 36 [6], neumoparotiditis [10, 76], pneumoparotitida [76], pneumoparotite [43], pneumopa- 37 rotide [55, 93], and parotisemphysem [47].”
- In the introduction it is explained that there is already a recent review. It is not clear then what this new review will add. Please clarify the objective of this review (line 60-61)
Methods
- Please add “(PRISMA)” (line 64-65)
- Please, share the search strategy, not only the words (line 68-69)
- Please, explain how many researchers assessed the references.
- Please, correct “redisposing” to “predisposing” (line 95)
Results:
- Please, in the methods explain what is the definition of disease duration (line 114). Is the length of the illness until complete resolution of the parotid swelling episodes?
- I’m not sure if sialendoscopy could be considered a diagnostic image. I would remove it from the table. Same in line 131. I find interesting using sialendoscopy in pneumoparotid, so I would include a different paragraph to summarize the evidence regarding sialendoscopy findings.
- Figure 2 and Figure 3: are these images from the author? If not, they should be referenced.
- In line 187-188 it is stated that “The number of patients continues to increase”, but with this data we only can say that the number of reported cases has increased.
- Please, change “follow-upin” to “follow-up in” (line 213)
Discussion
- It is unnecessarily large. Please, make it shorter.
- Antibiotic are prescribed (line 198), its role should be discussed. Why? What is the evidence?
- Antiinflamatory drugs are prescribed (line 199), it should be discussed. Why? What is the evidence?
- What is the role of sialendoscopy as therapeutic method? (line 379-382)
- Please, add in limitations the huge missing data.
Author Response
Introduction
- Too many references (line 33-34). I would eliminate all this “…such as pneumoparotitis [15, 17, 21, 25-29, 31-34, 36, 37, 39, 45, 48, 49, 33 53, 54, 57, 59, 61, 63, 64, 66-68, 72, 78, 82–84, 86, 88, 90, 95, 100, 103, 104, 106, 109], pneu- 34 moparotiditis [12], pneumoparotitis [29, 64, 101], pneumatozele [1, 4, 128], pneumosiala- 35 dentitis [24], pneumatocele glandulae parotitis [9], wind parotitis [13], pneumoparotidite 36 [6], neumoparotiditis [10, 76], pneumoparotitida [76], pneumoparotite [43], pneumopa- 37 rotide [55, 93], and parotisemphysem [47].”
Thank you very much for reviewing my manuscript. I am most grateful for your constructive and helpful comments on my manuscript.
As a recent review (Gazia et al. 2020) pointed out, the terms pneumoparotitis or pneumopatotid are often confusing or misused. I thought I should mention the current situation where the terminology is not unified. I listed the terms in order to emphasize that the terms are extremely diverse and not unified. Indeed, the sentence is very long with too many references. However, I don't think this sentence should be deleted. I limited English terms only and revised the sentence as follows;
“Various English-language terms have been used for this condition, such as pneumoparotitis [15, 17, 21, 25–29, 31–34, 36, 37, 39, 45, 48, 49, 53, 54, 57, 59, 61, 63, 64, 66–68, 72, 78, 82–84, 86, 88, 90, 95, 100, 103, 104, 106, 109], pneumoparotiditis [12], pneumoparotis [29, 64, 101], pneumosialadentitis [24], pneumatocele glandulae parotitis [9], wind parotitis [13].”
- In the introduction it is explained that there is already a recent review. It is not clear then what this new review will add. Please clarify the objective of this review (line 60-61)
A recent review by Gazia et al. (2020) evaluated 54 patients from 49 English reports from 1987 to 2019, so they analyzed only 54 cases from 49 papers. I believe that data on rare diseases such as pneumoparotid should be collected as many cases as possible. Therefore, in this review, I analyzed the clinical data of 170 cases from 126 articles, without limiting the period or language. I revised following sentences in the Introduction.
“A recent review on pneumoparotid and pneumoparotitis evaluated 54 patients from 49 English reports from 1987 to 2019 [120]. Due to this condition's rarity, many cases might not have been correctly diagnosed and adequately treated. Therefore, identifying this entity is necessary through a thorough understanding of previously reported cases. On rare conditions such as pneumoparotid, data should be collected from as many cases as possible. Therefore, this study assessed and reviewed all reports in all languages on pneumoparotid, including clinical presentations and treatments.”
Methods
- Please add “(PRISMA)” (line 64-65)
I added the abbreviation “(PRISMA)” after the 2020 Preferred Reporting Items for Systematic Reviews and Meta-Analyses.
- Please, share the search strategy, not only the words (line 68-69)
Thank you for a valuable comment. The text has been revised as follows.
“The literature search strategy was based on comprehensive electronic medical literature databases (PubMed, Scopus, Web of Science, EBSCO, Ovid, Google Scholar, SciELO, LILIACS, Japan Medical Abstracts Society, CiNii, and J-Stage) for the keywords “pneumoparotid,” “pneumoparotitis,” ”pneumoparotitis,” and “pneumoparotide.” Furthermore, a manual search was conducted for articles on pneumoparotid or pneumoparotitis. Reports from 1890 until June 30, 2022, identified in these databases or via a manual search were screened by the author, with no language restrictions. Duplicate reports and double publications from the same cases were excluded. Exclusion criteria were records with missing fundamental information, such as sex, age, etiology, and clinical presentation, and those irrelevant to the purpose of this study., Further, all reports were assessed for eligibility and reviewed by the author. Reports on the "puffed-cheek" maneuver during computed tomography (CT) were excluded as they were irrelevant to the study objective. Reports of anesthesia mumps were included only when air was confirmed in the swelling. No restrictions were imposed with respect to the original text type.”
- Please, explain how many researchers assessed the references.
I, the author, screened, assessed, and reviewed all the references by myself.
- Please, correct “redisposing” to “predisposing” (line 95)
I didn't notice it at all. I corrected the mistake. Thank you for your detailed review.
Results:
- Please, in the methods explain what is the definition of disease duration (line 114). Is the length of the illness until complete resolution of the parotid swelling episodes?
I defined disease duration as the time from onset of pneumoparotid to the first visit of each report. I didn't write about it, so I added it to Methods as follows.
“The disease duration was defined as the time from the onset of pneumoparotid to the first visit reported in each study.”
- I’m not sure if sialendoscopy could be considered a diagnostic image. I would remove it from the table. Same in line 131. I find interesting using sialendoscopy in pneumoparotid, so I would include a different paragraph to summarize the evidence regarding sialendoscopy findings.
The application of sialendoscopy to pneumoparotitis has increased in recent years [Li et al. (2012), Konstantinidis et al. (2014), Goates et al. (2018), Ambrosino et al. (2019), Goncalves et al. (2022)]. However, the number of cases of pneumoparotid is very small, and unfortunately there are no studies with evidence. I discussed in the Discussion as follows;
“Sialendoscopy is a routine diagnostic and minimally invasive therapeutic procedure that aims to evaluate and manage salivary ductal system disorders, including chronic inflammatory conditions [95]. The use of sialendoscopy in the diagnosis and screening of pneumoparotitis has recently increased [87, 95, 108, 113, 126].”
- Figure 2 and Figure 3: are these images from the author? If not, they should be referenced.
Figure 2 and Figure 3 are photos of my patient. These photos have never been published previously.
- In line 187-188 it is stated that “The number of patients continues to increase”, but with this data we only can say that the number of reported cases has increased.
That's certainly true, so I rewrote it as you pointed out.
- Please, change “follow-upin” to “follow-up in” (line 213)
I didn't notice it at all. I corrected the mistake. Thank you for your detailed review.
Discussion
- It is unnecessarily large. Please, make it shorter.
As this is a comprehensive review, the text is really very long. I made my manuscript as short as possible.
- Antibiotic are prescribed (line 198), its role should be discussed. Why? What is the evidence?
I think that many authors would prescribe antibiotics if the patient had inflammatory findings, just as they did for parotitis. However, the number of cases of pneumoparotid is small, and unfortunately there are no studies with evidence.
- Antiinflamatory drugs are prescribed (line 199), it should be discussed. Why? What is the evidence?
I think it's exactly the same situation with antibiotics. Please refer the answer above.
- What is the role of sialendoscopy as therapeutic method? (line 379-382)
Sialendoscopy is one of minimally invasive therapeutic procedure that aims to evaluate and manage salivary ductal system disorders, including chronic inflammatory conditions. However, the role of sialendoscopy as therapeutic method for pneumoparotitis remains uncertain.
- Please, add in limitations the huge missing data.
Thank you for a valuable comment. I revised the text and added revised sentences in the Discussion as follows;
“Since previous reports concerning pneumoparotid lacked available data, the author could not perform common analyses, such as synthesis of results or meta-analyses. In this report, to clarify the differences between the two categories, isolated acute events and recurrent injuries, the results were divided into two categories and statistically analyzed; however, data were missing, preventing a reliable statistical outcome. Authors of case reports or case series should provide fundamental data that are amenable to evaluation in later systematic reviews or meta-analyses.”

Reviewer 2 Report
it looks like a comprehensive review on the topic of pneumoparotitis and it is certainly of interest. Maybe an extra table summarizing the important points of the sections on the diagnosis and management of pneumoparotitis would be useful for the "busy" clinician
Author Response
it looks like a comprehensive review on the topic of pneumoparotitis and it is certainly of interest. Maybe an extra table summarizing the important points of the sections on the diagnosis and management of pneumoparotitis would be useful for the "busy" clinician
Thank you very much for reviewing my manuscript. I am most grateful for your constructive and helpful comment on my manuscript. As you pointed out, Table S1 is very large, and I am fully aware that it is difficult for "busy" clinicians to read. Therefore, I attempted to devise ways to help the readers understand the outline of the text, tables, and figures. This manuscript is already very long and has been pointed out by other reviewers to be shortened. I fear that creating an additional table summarizing the main points will make the manuscript longer and more complex. A “busy” clinician will be able to understand this review after reading the text, figures and tables. If less busy readers are interested in more detailed data from each report, I suggest they can access Table S1.
